# Relationship Aspects of Mothers and Their Adolescents with Intellectual Disability as Expressed through the Joint Painting Procedure

**DOI:** 10.3390/children9060922

**Published:** 2022-06-20

**Authors:** Tami Gavron, Rinat Feniger-Schaal, Adi Peretz

**Affiliations:** 1Department of Art Therapy, Faculty of Social Sciences & Humanities, Tel Hai College, Upper Galilee, Qiryat Shemona 1220800, Israel; 2School of Creative Arts Therapies, Faculty of Social Welfare and Health Sciences, University of Haifa, Haifa 3498838, Israel; rinatfen@gmail.com (R.F.-S.); adi.peretz141@gmail.com (A.P.); 3Drama & Health Science Lab, University of Haifa, Haifa 3498838, Israel; 4Center for the Study of Child Development, University of Haifa, Haifa 3498838, Israel

**Keywords:** intellectual disability, adolescents, mother–adolescent relationship, joint painting procedure, art therapy

## Abstract

The quality of the interaction between mothers and their children with an Intellectual Disability (ID) plays a crucial role in their development and in particular during adolescence. This qualitative study was designed to provide a better understanding of aspects of the relationships between mothers and their adolescents with ID through an art-based tool, the Joint Painting Procedure. The qualitative analysis of six dyads of mothers and adolescents with severe, moderate and mild ID was based on the principles of narrative and phenomenological inquiry. The findings yielded three key themes that emerged from the relational dynamics during the JPP: (1) from dependency to autonomy, (2) the joint painting as a way to foster verbal communication, and (3) playfulness and enjoyment. The JPP appeared to serve as a meaningful art-based assessment of the implicit and explicit aspects of the relationships which evolved during the interaction. The findings underscore the potential of the JPP as a non-verbal, art-based tool that allows researchers and clinicians to learn more about the dynamics of relationships between mothers and their adolescents with ID. It also enables a context where the expression of relational issues can be communicated and even transformed.

## 1. Introduction

Intellectual disability (ID) manifests before the age of 22 and is characterized by significant limitations in intellectual functioning (IQ < 70) and adaptive behavior compared to peers from the same background and community [1,2]. In recent years, there has been growing acknowledgment that the functioning of children and adolescents with ID is affected by interactions between personality traits, genetic factors, family relationships and parental approaches, much like typically developing children [3]. Similarly, the quality of the interaction between mothers and their children with ID plays a crucial role in their development [4,5]. The phase of adolescence presents even more challenges for mothers and their adolescents with ID [6]. The present study focused on better understanding the relationship between mothers and their adolescents with ID using the Joint Painting technique, a non-verbal, art-based tool.

Parenting a child with ID is an immense challenge that can impact parenthood and the mother–child relationship [3,7]. Children with ID have multiple difficulties that can affect their relationships with their parents, since they experience not only cognitive difficulties, but also difficulties in their emotional and social understanding [2]. Children with ID often display little initiative [8] and limited responsiveness [9]. Consequently, parental interactions with these children is perceived as less rewarding, and is characterized by less synchronization, pleasure, and reciprocity than for parents of typically developing children [10]. More specifically, studies have found that mothers of children with ID tend to present a more didactive parenting style [11,12], and often use controlling and intrusive behavior [13]. In addition, mothers of children with ID report feelings of tension and guilt [14], stress [15] and a need for support [16].

The quality of parent–child interactions is a significant marker of present and future child and adolescent development. In contrast to the considerable research on typically developing children [17], few studies have examined the interaction between parents and their children with ID. Within this literature, most studies have focused on a specific aspect of parenting such as structuring [18] or comforting [19], and have primarily dealt with samples of young children with ID (for example [17,20]). 

The phase of adolescence, which includes complex physical, social and emotional changes [21], may present an even greater challenge for the mother–adolescent relationship [22]. Adolescents with ID and their families are predisposed to a variety of behavioral problems and often need to cope with bullying [23,24]. Studies comparing adolescents with and without ID have noted that the former experience a higher rate of psychopathology, a lower level of satisfaction, greater symptoms of depression and increased bullying [6,25]. In addition, mothers of adolescents with an intellectual disability often report higher levels of parenting stress than parents of typically developing adolescents [22].

Although there is extensive research on the influence of the mother–child relationship on the emotional world of typically developing adolescents [21,26], the relationship of mothers–adolescents with ID has not been sufficiently explored [4]. Most research has focused on relations during childhood or adulthood and has dealt with the parental experience and its impact on the characteristics of the child’s behavior [27,28,29]. Studies conducted with mothers and adolescents with ID relate mostly to the motherhood experience of raising an adolescent with ID [30,31,32]. For example, studies indicate that mothers of adolescents with ID are concerned about their adolescent’s future and the need to use specific coping strategies [33].

Assessing the relationships between mothers and their adolescents with ID is complex, since these adolescents have difficulties verbally expressing their perceptions of the relationship [34]. Most studies rely on verbal reports, but human relationships are composed of two channels of communication: an explicit channel which is captured by the verbal account, and the implicit channel that is connected to non-verbal, procedural and nonconscious expression. These two dimensions develop in tandem, suggesting that relationships should be evaluated on both these levels [35]. Researchers who have probed the implicit aspects of relationships indicate that to evaluate implicit aspects, they should ideally be examined during an interaction [35,36]. One way to evaluate these implicit qualities of the relationship between parents and children is through joint creative processes [37,38]. Over the last four decades, extensive clinical literature has pointed to the value of joint painting and drawings as a way to better understand family relationships [39,40]. These instruments contribute to the evaluation of the implicit dimensions of relationships but have rarely been empirically tested [41].

The present study examined aspects of implicit relationships between parents and adolescents with ID using an art-based assessment, which can thus bypass the verbal obstacles characterizing population with ID [34,42]. It implemented the Joint Painting Procedure (JPP) [43], a validated, art-based assessment tool that has been used in research and clinical settings for over a decade with children in middle childhood [37,44,45].

Joint parent–child painting can shed light on the implicit representations of each partner, as well as the shared representation of a specific dyad. A shared relationship representation differs from an individual representation in that it relates to the behavior of each member of the dyad toward the other and can capture mutual emotional responses in the “here and now” of the mother–child interaction. This ongoing interaction can provide a clearer picture of the implicit aspects of the actual relationship [37,45]. Previous studies using the JPP indicate that it allows for the evaluation of a range of aspects of the implicit relationship conveyed nonverbally through the painting, the artistic process and the behavior [37]. To the best of our knowledge, no studies have examined aspects of the implicit relationship between mothers and their adolescent children with ID. This study was designed to fill this gap by examining the characteristics of mothers and adolescents with ID nonverbally during a “here and now” interaction using the JPP.

## 2. Method

This study is part of larger mixed-method research that explores the parent-child relationship of mothers and adolescents with ID. It presents the qualitative part of the study that sheds light on their relationship.

### 2.1. Participants

The participants were on a list of families who took part in a study conducted 11 years ago on families with children with ID, who were originally recruited through child development clinics. For the purposes of this study, the families were approached by letter followed by a phone call to ask whether they would be willing to participate in the study. The aims and procedure were explained and a meeting was scheduled at the family’s home.

This article describes six dyads of mothers and adolescents (five girls and one boy) who were representative of the levels of disabilities of the broader study population; namely, two adolescents with severe disability, two adolescents with moderate disability and two adolescents with mild disability. All the participants were diagnosed as having a non-specific ID (with unknown etiology). The IQ of the adolescents ranged from 40 to 78, based on the Stanford–Binet intelligence scales [46] that was administered to all the participants as part of the larger study. The mothers ranged in age from 46 to 58, and the adolescents ranged in age from 16 to 20. All the mothers were Israelis. One mother was Druze. One Jewish mother self-defined as Orthodox whereas the others stated that they were secular. The mothers’ years of education ranged from 11 to 15 with a mean of 12.6. One dyad’s native language was Arabic, whereas the others were native Hebrew speakers. All names used in the text are pseudonyms to preserve the participants’ anonymity.

### 2.2. Procedure

The JPP administrator (the third author) met with the dyads in their homes for two to three hours. After a short interaction with the dyad and a brief explanation of the procedure, the researcher and the mother found a suitable space for the joint painting process. Then, the mothers and the adolescents engaged in the JPP, using materials that were provided by the administrator. The procedure was videotaped and still photographs of the artwork were taken. The administrator took detailed notes on the implicit and explicit aspects of the interaction (verbal, behavior and affective components) in each phase of the JPP. A self-reflection diary was written by the researcher/administrator after each home visit. The researcher met the second author (the developer of the JPP) after each session to view the video and discuss it.

This study was approved by the Ethics Committee of the University of Haifa (# 18/036).

### 2.3. Assessment Tool

The Joint Painting Procedure—JPP [43]. The JPP is comprised of a structured five-step process in which both partners paint on the same sheet of paper, first working separately side by side, and then painting together on a shared area. In the first step, the mother and child are asked to use a pencil to mark their own personal space on the shared sheet of paper. Next, each partner paints inside his or her personal space using gouache. This is followed by the instruction to draw a frame around the painted space, and then paint a path from that frame to the frame painted by the partner. In the fifth and final step, parent and child are asked to paint the rest of the paper together. After painting, the parent and child look at the painting with the therapist, discuss the shared experience, tell a story about the painting and give the painting a title.

The JPP, which is grounded in the principles of parent–child art psychotherapy [47] and art-based assessments [41], is based on the assumption that diagnostic information is embodied in the way in which the artwork is done, beyond the symbolic content in the artwork. The emphasis is on how people paint and not just about what they paint. The JPP analysis assesses the formal elements of the joint painting (i.e., color, forms, shapes), and assumes that these elements provide information about various implicit aspects of the relationship. At the same time, it incorporates symbolic content such as images and metaphors [43]. The JPP has a validated coding manual designed for middle childhood that includes seven scales, with descriptions of the phenomena that characterize each level of every scale, and relate to the painting process and the final product, as well as behavioral phenomena in each stage of the process. During the data collection phase (administering the JPP to the dyads) which took 12 months, the manual was adapted to adolescents with ID by the researchers including the inventor of the JPP (the first author). In this study, the qualitative analysis of the joint painting used the scales as a way to observe phenomenological phenomena for the purposes of structuring the observation and was not intended to code it numerically [45].

### 2.4. Data Analysis

The qualitative analysis was based on the principles of narrative and phenomenological inquiry [48,49]. The aim was to conduct an in-depth observation of the ways in which the joint painting process evolves as a reflection of the narrative of the dyadic relationship [50,51]. This phenomenological approach served to investigate the nature of the mothers’ and adolescents’ relationship as revealed in the process and product of the joint painting [48,52]. The researchers analyzed the concrete descriptions (phenomena) produced in the artwork (i.e., shapes, forms, colors, images) [49]. The narrative inquiry situated the dyad within a storyline where the evolving narrative was composed of the dynamics of the dyadic relationship as conveyed through the process and product of art-making [53]. The dyadic interactions were observed in their natural context in the participants’ home and examined the way the relationship unfolded and changed during the joint painting [49,54]. The narrative qualitative inquiry in this study focused on the narrative as it unfolds during the interaction through verbal communication, behavior (individual and joint), the evolvement of the painting process and the narrative embedded the painting itself (analyzed from a phenomenological perspective). The analysis sheds light on the complexity of the dyads’ experiences throughout all the phases of the process [49].

The analysis took place in three stages: 1. observing the process separately for each dyad; 2. creating and writing an inclusive narrative for each dyad based on the joint painting, the artwork and the dyad’s behavior; 3. consolidating the insights for each dyad and integrating the explicit and implicit aspects of the relationships as they were expressed during the painting process and in the artwork into main themes and sub-themes. This was done by comparing cases to identify the central themes, similarities and differences [55]. 4. Then, the researchers examined and discussed each theme in relation to each dyad.

To strengthen reliability, a number of analysis methods were implemented [56]. The authors examined the videotaped interaction as well as the paintings of each dyad, and made considerable use of the written self-reflection and in-depth descriptions of the third author. The third author conducted on-going consultation sessions with the research facilitators (authors A and B) for the purpose of triangulating points of view.

## 3. Findings

The narrative analysis was based on total of six dyads, four of which are described in depth below. All six dyads are described in the themes section.


*Case Study # 1: Rona (17) with severe ID (IQ < 40) and her mother.*


Rona looked excited before the joint activity, and she smiled and hugged her mother. Her mother stroked her head and said: “Come on Rona, we have work to do”. Rona held her mother’s hand and her mother gave her the pencil and said that they both had pencils. Rona’s mother outlined the shape of a heart and Rona scribbled across the page and tapped on it with the paintbrush, with excitement. When the mother realized that Rona could not draw the outline of the shape, she held Rona’s hand and drew a heart with her and said: “Mom drew a big heart and Rona is drawing a small heart”. Rona tapped her mother’s shoulder with a broad smile.

When they were asked to draw in their individual spaces, Rona’s mother placed a paintbrush in Rona’s hand and said: “A paintbrush for you and one for me”. Rona was curious about the paintbrush and painted a blue line without any help. Her mother encouraged her, and Rona tapped on the page with the paintbrush. Her mother outlined the heart she had drawn in red and connected it to Rona’s border. When her mother encouraged Rona to continue, Rona painted in black inside and outside her mother’s heart, and then added yellow inside her mother’s heart shape. Rona mixed a large quantity of different color paints until the color turned brown–purple, and then smeared thick layers on the page. Her mother painted the heart that she had drawn beforehand for Rona using the thick paint that Rona had left on the page, and Rona painted over it in various colors (see Figure 1).

Rona became tired, placed the paintbrush to the side and rested her head on her mother’s shoulder. Her mother put another paintbrush in Rona’s hand and tried to help her find a color, but Rona refused, and painted with her finger on her hand. Her mother put her hand over Rona’s hand and helped her paint with her hand on the page, but Rona preferred to continue painting on her hand and not on the page. Her mother painted on the page using her finger and said to Rona: “Now it’s your turn”, but Rona still refused to join her.

When asked to draw a border around their paintings, Rona’s mother again drew Rona’s attention to the big heart frame in turquoise. She asked Rona: “Where is your heart?” but Rona did not react and refused to draw a border, while continuing to paint on her hand. Her mother painted a pink heart outline for Rona. Rona was excited, took the paintbrush from her mother’s hand, dipped it into her mother’s palette and together they reinforced the border of Rona’s heart in turquoise. When asked to draw a path, Rona’s mother painted a pink line from the inside of her heart to the inside of Rona’s heart and continued to thicken the borders of the hearts in the same color. Rona did not want to paint a path and continued to paint on her hand. In the joint space, her mother painted orange and blue scattered circles and one small heart inside her heart. She tried to paint dots in the style of Rona’s tapping and then tried to hold Rona’s hand and paint with her, but Rona refused and continued painting on her hand.

In the discussion after the painting process, Rona’s mother mentioned that the colors were beautiful and asked Rona what they should call the painting. In response, Rona looked at the painting, hugged her and held her hand. Her mother said that they could call the artwork “Hand in Hand” and she told a story: “Rona and I spend a lot of time together, we walk hand in hand, we take walks together, and this is our story, hand in hand”.

To summarize: During the entire painting process, Rona’s mother tried to help Rona and be attentive to her needs, while trying to be ‘with’ her in the painting by painting in a similar way. For example, she imitated Rona’s style and choice of colors when finger painting. Rona’s mother commented verbally on their joint work and their similarities. While supporting Rona, she was able to express herself in the painting. During the process, Rona sought her mother’s touch, and it seemed to help her regulate her emotions. She got attention and reinforcement, and as a result was often able to create without assistance and express herself in various ways (i.e., the way she chose and mixed colors), and seemed to have the choice to withdraw when she was tired. In addition, the painting process and the product seemed to display processes of both autonomy and closeness. This occurred, for example, when her mother repeatedly bolded the outline of both heart shapes, or when Rona and her mother painted freely in each other’s shapes (see Figure 1).

Naama appeared timid and embarrassed, and she clung to her mother. Her mother drew an ellipse on the right across the width of the page, and Naama imitated the same shape in a smaller form on the left. When they were asked to paint in their own spaces, Naama said she was falling asleep and hugged her mother. Her voice was weak and bashful. Her mother colored the ellipse turquoise and encouraged Naama to join her, and then Naama colored her ellipse in turquoise using small brushstrokes.

When asked to draw a frame around their spaces, Naama’s mother painted a red frame attached to the ellipse, and Naama painted a thick purple frame and described what she was doing. When asked to paint paths, her mother painted a full, thick yellow line that began at Naama’s frame, and then turned to Naama and said: “Your turn”. Naama drew a green line parallel to her mother’s path, starting from mother’s frame to hers. The line, however, did not touch either end of their frames.


*Case Study #2: Naama (17) with moderate ID (IQ = 44)*
*and her mother.*


When asked to paint in the joint space, her mother asked Naama what and how she wanted to paint. Naama suggested filling in the areas with color and asked her mother to paint a yellow heart which she then colored purple. Her mother suggested that this time Naama should draw something, and she would paint inside it, but in response Naama raised her voice slightly, refused, and then drew a small orange shape next to her own space. Her mother reinforced it and filled the shape with light blue. Naama’s mother then suggested that each of them could paint a sun and that Naama could paint in other places on the page. She asked Naama to choose what to paint, but Naama admitted that it was hard for her to decide. With considerable mediation by her mother, Naama painted a light blue sky. Her mother asked if she could continue the sky on her side of the page in a different shade of blue, and Naama answered “Only use light blue” and pointed directly at where she could paint. They both painted parallel skies, in each of their own spaces, in the same painting style, and Naama connected them. Naama’s mother added white clouds and black birds and explained they were like the birds that fly over their house. Naama painted blue birds, and her mother reassured her and asked what else she could add by pointing to the remaining empty space on the page. She asked, “If there is a sky up above, what can we paint below?” Naama answered “The ground”. Her mother suggested they paint the ground on both sides connecting them. Her mother added a turquoise pool and Naama added a blue and black pool. Her mother encouraged her to go on painting and suggested she could add the figure of a child. Naama responded enthusiastically and painted an orange- and grey-colored figure of a boy and asked her mother to draw a girl. Mother painted a girl with an orange head and asked Naama humorously to give her some positive feedback. In response Naama hugged and kissed her. Naama painted another sun, but this time in the sky and not on the ground. Her mother painted a tree with lemons, a beetle and a snail, and Naama painted grass and a snail (see Figure 2).

When they were asked to give the painting a title, Naama’s mother insisted that they should decide together. Naama suggested “Rain” and they both said that the story was about a boy and a girl who go for a walk. Naama asked if she could take the painting to her bedroom.

To summarize, during the painting activity, a process of change was apparent. At first, Naama seemed embarrassed and painted hesitantly with small brushstrokes, slowly, and by focusing on a small shape. Her mother, by contrast, painted with flowing movements and spoke liberally. When connecting their individual spaces with paths, Naama seemed to reach to the dominant path her mother painted along with her comment “Your turn”. Her painting style then seemed to change slightly and displayed involvement and confidence. This persisted into the joint painting, during which Naama actively responded to her mother’s request to make decisions. She gave instructions, expressed herself from time to time, and even raised her voice somewhat when she was not pleased.

Most of the interaction between Naama and her mother, as expressed in the joined painting, was based on her mother’s attempt to mediate the painting process, encourage her to paint, help her make choices, and support her during painting. Her mother tried to be attentive to Naama and her needs, by upholding and providing her with a space to express herself on her own. The joint painting appeared to have provided Naama with a secure space to express herself and allowed both mother and daughter to experience mutual and shared creative expression (see Figure 2).


*Case Study #3: Lianne (17) with moderate ID (IQ = 57) and her mother.*


Lianne sat down immediately to the right side of her mother, and seemed excited. She wanted to start painting right away and was the first to paint on the page, creating a square frame with a heart shape inside and asking her mother (who had not started to paint yet) “Who paints better?” Her mother appeared tenser and more hesitant, and painted a square frame with a moon shape inside. When they were asked to paint in their individual spaces, Lianne painted a red heart in quick short brushstrokes. While she was painting, Lianne suddenly remarked loudly: “Mom, we never paint together, only when I was little, and that was a long time ago”. Her mother agreed and complimented Lianne on her painting. Lianne suggested that her mother paint with turquoise and white, and her mother responded to her suggestion. Lianne’s mother painted slowly and did not interact with Lianne at this stage. Lianne mixed colors on the page and her mother suddenly commented in a relatively loud voice that Lianne’s mixture produced a lovely color.

When they were asked to paint a frame, Lianne painted intensively and made red strokes with spaces, and then added white-turquoise strokes until the border turned pink. Her mother painted a pink frame with long slow movements. She commented that Lianne had painted a special frame and Lianne answered with pride: “I was born talented”. When asked to paint paths, Lianne said she would choose black, and then burst out laughing. She said: “A black path!” and laughed again, and her mother answered that she could do whatever she wanted. Lihy painted a pink path from her frame towards Lianne’s with delicate brushstrokes. Lianne painted a turquoise path from her mother’s frame to her own, and then added a parallel purple path but in the opposite direction. Lianne’s mother then reinforced her pink path and Lianne imitated her. Then, her mother asked if the paths needed to be joined and immediately Lianne connected them at the point that touched her frame and painted onto a small section of her mother’s path (see Figure 3).

When asked to paint in the joint space, they engaged in a discussion that led to joint artwork: Lianne’s mother asked her what she would like to paint and told her that she was creative. Lianne suggested they write their names in English and they both wrote their names in pink. Lianne commented that people tell her she has nice English handwriting, and then she bolded her name. Her mother responded in a soft voice that she writes beautifully. Lianne and her mother decided to paint the shape of a heart around their names together. Lianne said she was excited and they both smiled. Her mother indicated clearly that she was going to paint on her side in red and started painting, while Lianne painted another part of the heart in pink and made it thicker. Lianne said excitedly: “Everything is going well for me today; it’s like a miracle”. Finally, Lianne added purple stripes inside the heart, and her mother called this addition a “decoration” and complimented her. When they finished painting, Lianne commented that the artwork was very beautiful, hugged her mother and called out “my mom”. Her mother suggested they title the painting “Creative Lianne”. Lianne said that the painting was about “the way to the moon leads to the heart; you’ll always aspire for the moon or to your heart” and decided to call the painting “The Way to the Moon leads to the Heart”. She said that she had enjoyed the joint part of the painting experience very much. Her mother said that she particularly liked when they completed the heart together, and that Lianne’s idea had helped her.

To summarize, Lianne may have been testing out separateness from her mother, which emerged in her sense of competitiveness and the comparison at the beginning of the painting process, as well as later on when Lianne said she was going to paint a “black path”. Nevertheless, while seeking autonomy, Lianne sought to be close to her mother by using similar colors and saying that she specifically enjoyed the joint painting and the moments when they connected. During the painting process, a change appeared to have occurred in the relationship. Her mother started from a more distant and hesitant stance. When asked to paint a path towards Lianne’s frame, she painted a thick, prominent line. Lianne responded to her mother’s path by connecting them both. From this stage onward, they both appeared to be enjoyed expressing themselves mutually and individually. The joint heart which was painted during the last stage of the process contained colors from their individual spaces that were associated with their names written inside the shape (see Figure 3). During the creative process, there was a noticeable change in the mother’s behavior, from a hesitant presence to active involvement and positive attributions about Lianne. Lianne’s story metaphorically described a journey towards closeness.


*Case Study # 4: Meital (16) with mild ID (IQ = 70) and her mother.*


Meital and her mother looked relaxed and started painting together. Meital painted a square frame and her mother painted a circle. When asked to paint in their individual spaces, Meital’s mother said that she did not know what color to choose, and Meital reacted angrily by saying that her mother had never told her what her favorite color was. Meital’s mother painted inside the circle in various colors and Meital painted in black. They were both quiet and careful not to color outside of the frame. Meital’s mother said: “I have not painted for a long time” and Meital answered furiously: “A long time, never!”

When asked to paint the frame, Meital painted a wavy purple border, and her mother painted a round red frame with black dots. Meital told her it was pretty, and her mother giggled and said: “Thank you, this is fun!”

When they were asked to paint a path, Meital’s mother suggested painting two paths that would meet. Meital agreed enthusiastically and started painting in green. Her mother guided her on how to paint and how to hold the paintbrush, although Meital did not appear to have any problems doing so. They painted in green together, towards each other, and met in the middle. Meital’s mother made the paths thicker and asked Meital if she wanted to add any extra decorations. Meital decided that they would paint flowers below the path and together they decided to add more paths: her mother painted a brown path underneath the joint path, and Meital painted an orange path above it towards her mother (see Figure 4).

During the joint painting stage, Meital’s mother asked her what she would like to paint, and Meital answered: “Whatever you like, mom”, and her mother said “So we are not deciding on something together? What should we paint and where?” Meital suggested they paint butterflies, hearts and trees. Her mother painted two connected hearts in blue and turquoise, and Meital painted two butterflies. Her mother painted two more hearts below Meital’s drawing, and then commented twice: “I am invading your drawing space,” and added “you are welcome into my space.” Meital painted a turquoise heart shape, and her mother painted a colorful butterfly. When her mother commented that there was more empty space on the page, Meital suggested that they each paint a cloud. Meital’s mother asked her if she knows how to make grey, and Meital said that did not know. Her mother explained to her and demonstrated how to mix the colors together. They both added rain drops, Meital in blue and her mother in turquoise. Meital’s mother then asked what else they could paint, and Meital replied that they could paint a sun. Her mother suggested that Meital paint the sun, and Meital immediately responded that this was a joint painting, and that they could do whatever they liked. Finally, Meital painted a yellow sun on her side of the page (see Figure 4).

When they finished painting, Meital suggested giving the painting a title that related to both of them. They chose the title “Mother’s and Meital’s painting”. They both liked the painting: Meital liked the butterflies and the connecting paths, and her mother liked the clouds and rain.

To summarize, during the creative process, the interaction between Meital and her mother shifted from a relaxed atmosphere to anger when Meital reacted to the fact that her mother had never told her what her favorite color was, and about not painting together. At this stage, Meital only painted in black, and later, when her mother painted a colorful frame, Meital responded positively and used other colors (see Figure 4). At this point the mood changed, and there was laughter in the room and both of their paintings became more colorful. This atmosphere continued during the joint painting stage, when the conversation between Meital and mother was respectful and mutual. This type of communication was also portrayed in the artwork that included pairs of similar figures such as the sun, butterflies, and hearts.

### 3.1. General Themes: Relationship Dynamics

The analysis of the six case studies yielded three key themes reflecting the dynamics of the relationship during the JPP: (1) from dependency to autonomy, (2) the joint painting as a way to foster verbal communication, and (3) playfulness and enjoyment.

#### 3.1.1. From Dependency to Autonomy

**Assistance versus autonomy**. The mothers apparently felt the need to help their daughters paint. At the same time, they acknowledged their children’s separate space that allowed the adolescents to express themselves as a function of their own abilities and needs. Some of the mothers found it difficult to encourage their daughters to create autonomous artwork. When the mothers enabled autonomy, the adolescents responded by more independent functioning, positive behavioral patterns, and fewer conflicts (as in the case of Meital and her mother # 4). By contrast, with Rona and her mother (case-study #1), the painting process elicited a tension between the need to support Rona and granting her autonomy and separateness.

Nevertheless, the tradeoff between the need to support and granting autonomy was present in all the dyads irrespective of the level of disability. For example, Shirin (IQ = 40) alternated between scribbling on the joint page while playing and enjoying herself, to playing with her mobile phone. Her mother spent most of the time encouraging, helping and re-engaging Shirin in the creative task. Her mother reported that it was difficult for her to cope with Shirin’s low functioning and her inability to follow instructions. However, at times, her mother failed to observe Shirin’s desire for autonomy; for example, when Shirin asked to paint in a specific color and her mother ignored her and handed her a different color. At a specific point during the painting process, her mother moved away from the page and Shirin began painting a few strokes on her own. It looked like they were playing a game of “tag” where every time her mother encouraged Shirin to paint, she avoided her, and when her mother moved away, Shirin began painting.

The issue of separateness emerged verbally during the painting process in a dyad composed of Ben (with mild ID) and his mother. Ben’s mother delineated a large space for herself that covered half the page and physically approached Ben’s side of the page. Ben complained that she was occupying a larger part of their page, and turned his back to her, painted in his space that was detached from the mother’s space, and commented that he was demarcating an area for himself. When asked to paint a frame, Ben painted a very thick one. During the entire painting process, Ben refused to paint together and furiously rejected his mother’s suggestions. Anger and a lack of willingness to cooperate may have represented his need for separation and autonomy from his mother.

**Enabling adolescents’ expression of different roles within the dyad**. During the joint painting, the adolescents were able to express themselves within the relationships verbally and implicitly and transition from a passive to a more active role. For example, at the beginning of the joint painting process, Meital’s mother (case-study #4) was dominant and tried to guide Meital by telling her what to do and how to do it, but as the process continued, her mother suggested that Meital make her own decisions about the painting and invited her to occupy a space on the page both verbally and through art. Meital reacted positively to her mother’s encouragement and started to be more active and involved in the painting process. She then was able to decide on the content and its place in their joint painting. Similarly, in Lianne and her mother’s joint painting (case-study # 3), the creative process became mutual and they could both express themselves. Lianne manifested her sense of success and satisfaction when she said that everything was working out like a miracle.

**A space for mutual recognition.** The joint painting appeared to provide an opportunity for these adolescents to express their inner world while remaining in the shared dyadic experience of joint and playful artwork with their mothers. For example, during Naama and her mother’s painting process (case-study #2), Naama’s mother tried to support Naama and at the same time encourage her independent expression. This allowed them to form a joint mutual space where they both painted the sky and the ground in their own spaces at their own pace and brush strokes until they met. Her mother expressed her desire for connection through her conspicuous path and by asking Naama to paint closer to her, while at the same time encouraging Naama to paint whatever she wanted. In the final painting, there were representations of two figures that were similar, but each has its own unique characteristics. At the beginning of Lianne and her mother’s painting process (case-study # 3), there was tension between Lianne’s dominance and her mother’s hesitancy. When the mother implicitly expressed her wish for a connection via her thick pink path, Lianne immediately responded, and a mutual creative dialogue was initiated while they painted the joint heart in which each contributed an individual expression.

#### 3.1.2. Painting as a Way to Foster Verbal Communication

Conversations emerged during the painting and afterwards, when the mothers and adolescents observed the final artwork together. For example, during Meital and her mother’s painting process (case-study # 4), Meital was able to express her anger and frustration. In the end, Meital’s title expressed their sense of togetherness. Naama (case-study # 2) was able to verbally express her difficulty deciding on the theme of the painting, but as the creative process unfolded, she was able to clearly express her ideas verbally. The discussion at the end allowed some of the dyads to reflect on the interaction while engaging in conversation where they could share their thoughts, feelings, and experiences about the process and the product. For example, Lianne and her mother (case-study # 3) told the story of a girl on a journey toward closeness that actually took place during the painting process.

#### 3.1.3. Playfulness and Enjoyment

For most of the dyads, the joint interaction had a didactic element that consisted of guided questions, instructions and teaching on the part of the mother, regardless of the level of disability of the child. However, during the creative process, the mothers and adolescents started to communicate with each other through painting, which enabled some of the dyads to create an atmosphere of playfulness and enjoyment. This playfulness was expressed through gestures, speech, and specifically, in the paintings themselves. For example, in the case of Shirin and her mother, in spite of the mother’s need to direct and help Shirin, and her concern about her daughter not following the instructions, Shirin expressed positive affect throughout the entire process by laughing, making ecstatic sounds, clapping her hands, forming eye contact and touch. Even though Shirin was often distracted and used her phone as a means of self-regulation, she managed to express herself in the painting in a rich way: it was full of thick colorful lines. Enjoyment was also evident in the creative interaction and exchange of humor between Naama and her mother (case-study #2). Similarly, Rona and her mother (case-study # 1) seemed to experience moments of enjoyment with spontaneous gestures. This was particularly evident when her mother playfully joined Rona in painting with her finger.

## 4. Discussion

The present study examined aspects of implicit relationships in mothers and adolescents with ID through a non-verbal art-based tool. The findings help reveal and facilitate a better understanding of the relationship dynamics that develop between mothers and adolescents with ID. The JPP was shown to provide a meaningful art-based assessment of the implicit and explicit aspects of the relationships that evolved during the interaction.

One of the central themes that emerged in all the dyads was the negotiation between dependency and autonomy. Autonomy is a central relational issue between parents and typically developing adolescents [57], but is also present in the relationships between parents and adolescents with ID [6]. This may be explained by the specific tension between dependency and autonomy in children with ID, where the dependence experienced by adolescents with ID may lead to compliance but also to difficulty in expressing anger and separateness, which in turn may lead to difficulty in becoming autonomous [32].

A previous study that examined processes of separateness and autonomy through the JPP in dyads of mothers and typically developing children found that autonomy is a crucial facet in the relationships of typically developing children [45]. In the present study, adolescents with ID allowed themselves to express their anger and separateness within the context of making art together. As the findings suggest, the JPP allowed for the expression of both dependency and autonomy. The joint painting allowed the adolescents to find their own separate voice. They could make decisions and occasionally control the situation, which is often impossible for adolescents with ID because of their disabilities [58,59]. Hence, the JPP acted as an assessment tool and as an intervention process that enabled transformative processes.

The joint painting process allowed the dyads to move between mutual experiences of “togetherness” to an individual, autonomous space. The process of mutual recognition was expressed through personal images or the painting styles of each partner that involved certain shared elements such as connected figures or a joint pictorial story. The painting process that developed from the individual work towards a joint work facilitated a rich relational process [37,43]. Mutual recognition within the relationship involves the ability to recognize the self and the other as having separate inner worlds, who at the same time can have a close mutual relationship [60,61]. Mutual recognition does not typically characterize interactions with children with ID [10]; however, in four dyads the painting process encouraged and enabled this recognition. In Rona’s and Shirin’s dyads with their mothers, the process of mutual recognition did not evolve. This could be related to their level of disability, their difficulties perceiving the point of view of the other, and their need for more assistance in comparison to the higher-functioning dyads [32].

Some of the parents presented a parenting style characterized in the literature as didactive or directive, which often characterizes parents of children with ID [11,12]. This parental behavior makes sense in view of the difficulties faced by children with ID and the support they need [62]. However, throughout the painting process, along with more guided assistance, the possibility for playful and creative mutual interactions emerged. Other studies that have examined the process of joint artwork of children and parents showed that creative work elicited feelings of enjoyment and playfulness and supported the relationship between the parent and the child [45].

As noted by Patton and colleagues (2018) [22], parenting children with ID leads to parental stress, which in turn can prompt negative parenting behaviors that lead to further difficulties in the relationship [63]. The joint painting process offers parents and adolescents a rare opportunity to engage in a creative, playful joint space, with no targeted end point and where they are encouraged to express their emotions and playfulness. As Lianne (case-study # 3) told her mother: “Mom, we never paint together, only when we were little, and that was a long time ago.” In a different study on the role of playfulness and joyful interaction in preschool children with ID [64], playful interactions with medical clowns had a significant effect on children’s development, whereas the parents of these children reported that they rarely managed to create a playful and enjoyable interaction with their children. The joint painting allows for a mutual experience and sense of enjoyment that seldom occurs in the daily lives of mother and an adolescent with ID.

## 5. Limitations

One notable strength of this study is the use of art-based assessment that served not only to learn more about the implicit relationships of mothers and adolescents with ID, but also to enable playful and creative communication, which is often rare. However, this study is not without limitations. This study used the JPP, a validated tool for parents and children in middle childhood. Although the manual was adapted for parents and adolescents with ID, further validation of the JPP for this population is needed. The present study examined the interaction between mothers and adolescents, but there are other meaningful relationships in the family, such as the relationship with the father, which was found to be essential for child development no less than the relationship with the mother [65]. Further research could examine the interaction between adolescents with ID and their fathers to explore implicit aspects of their relationship within the framework of joint painting. In addition, this study examined a very small number of participants with relatively diverse levels of disability. Future studies could utilize the JPP with a larger number of dyads that would shed more light on specific levels of disability. Research could also examine gender and cultural features that could impact the specific findings reported here.

## 6. Conclusions

Two main conclusions emerged from this study. The first relates to the importance of joint painting as a non-verbal art-based technique that allows researchers to learn more about the dynamics of relationships between mothers and adolescents with ID. The findings suggest that the joint painting process allowed for a deep and rich emotional expression on the part of each member of the dyad that yielded a representation of their dyadic relationship. The joint creative process and the shared observation of the product helped manifest a space of expression and communication that does not exist verbally because of these adolescents’ disabilities and the difficulties in parenting this population. Some of the joint paintings allowed for a process of change during the interaction where there was a shift towards a mutual space that afforded both mother and adolescent self-expression and enjoyment. From this perspective, the joint painting process provided an opportunity for a unique experience for these adolescents and their mothers.

The second conclusion is connected to the importance of the tradeoff between dependency and autonomy that emerged from the implicit expressions during the painting process. The inevitable tension that exists between the two is common in parent–adolescent relationships and in particular with mothers and adolescents with ID [59]. The JPP provided a context where the expression of these relational issues could be communicated non-verbally.

Learning about the inner world of adolescents with ID in general and their experience of their relationship with their mothers is challenging due to the verbal barrier. The present study employed an art- based method to learn more about the relationship between these adolescents and their mothers. The engagement and accessibility of the art activity for this population is a significant strength of the JPP, as was found in other art-based studies with adolescents, as well as with young people with ID (see for example, [66,67]. It enables direct access to the lived experience of these adolescents in the context of a relationship dynamic with their mothers.

The results point to the existence of a range of emotional features in the relationships of adolescents with ID and their mothers, much like in typically developing adolescents. It also shows that these implicit qualities can be changed and transformed through the painting process. As such, the present study provides valuable information that can be used by researchers and therapists who work with populations with ID and their parents. This study can inform therapists who work with this population on the potential of using an art-based dyadic intervention that is accessible, informative, engaging and possibly transformative for adolescents and their parents.

## Figures and Tables

**Figure 1 children-09-00922-f001:**
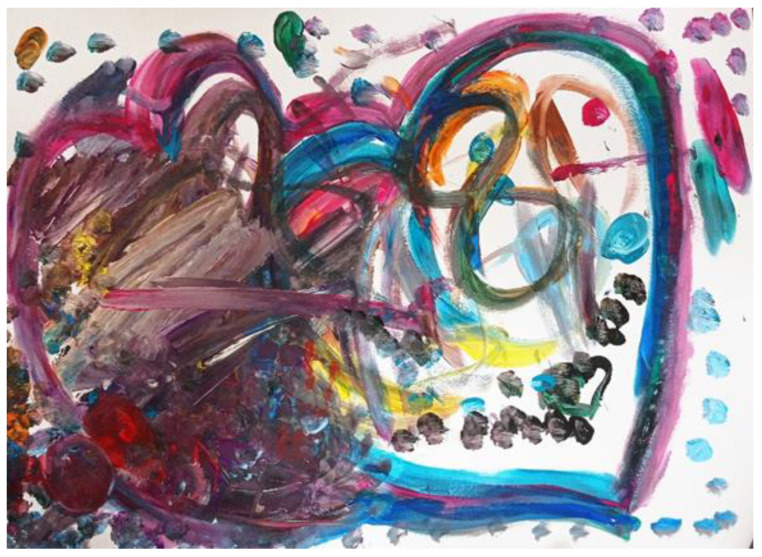
Rona and her mother.

**Figure 2 children-09-00922-f002:**
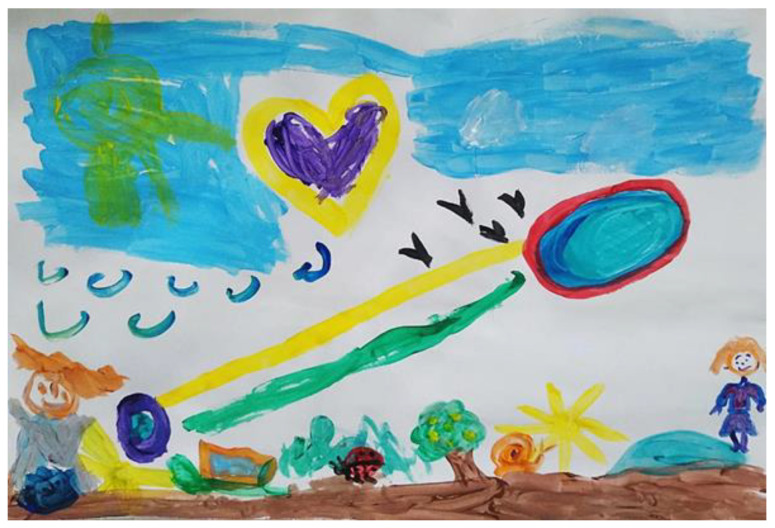
Naama and her mother.

**Figure 3 children-09-00922-f003:**
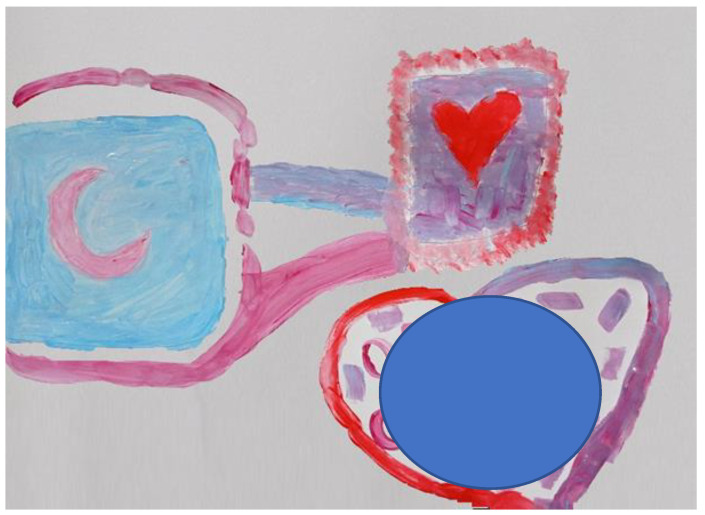
Lianne and her mother.

**Figure 4 children-09-00922-f004:**
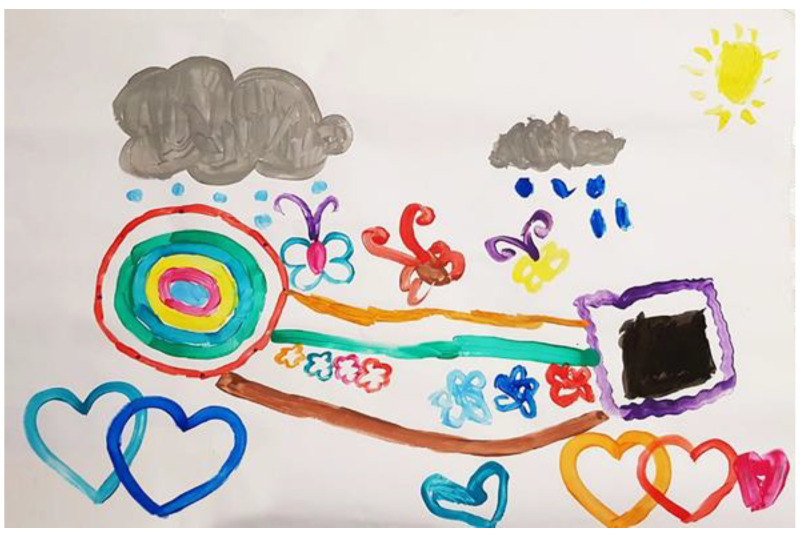
Meital and her mother.

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
