# Peer review of "Relationship Aspects of Mothers and Their Adolescents with Intellectual Disability as Expressed through the Joint Painting Procedure"

_children, 2022, doi:10.3390/children9060922_

Round 1

Reviewer 1 Report

The theme and the aim is  relevant and new and I really enjoyed reading it. But it could be improved for example:

-I think more detailed explanation of coding and analysis process is necessary.

-I think it could be interesting to show some sociodemographic data of mothers. Also,  the ethiology of the intellectual Disability of the participants.

-I guess that participants know the purpose of the paint. Could bias the mother's way of interactuate?

-Line 10: The new definition of ID has changed the limit of age (22 years instead of 18). The sentence presented in lines 155-156 could foster the infantilization of this population.  

Author Response

We thank you for your helpful and insightful comments and suggestions on the manuscript. The changes have been integrated into the revised version using track changes or are highlighted in color

We hope you will find the revised manuscript acceptable for publication.

Review 1:

The theme and the aim is relevant and new and I really enjoyed reading it. But it could be improved for example:

-I think more detailed explanation of coding and analysis process is necessary.

Please see page 5 lines 201-207:

The analysis took place in three stages: 1. Observing the process separately for each dyad; 2. Creating and writing an inclusive narrative for each dyad based on the joint painting, the artwork and the dyad's behavior; 3. Consolidating the insights for each dyad and integrating the explicit and implicit aspects of the relationships as they were expressed during the painting process and in the artwork into main themes and sub-themes. This was done by comparing the cases to identify the central themes, similarities and differences (Seawright & Gerring, 2008); 4. Then, the two researchers examined and discussed each theme in relation to each dyad.

- I think it could be interesting to show some sociodemographic data of mothers. Also,  the etiology of the intellectual Disability of the participants.

The participants were only identified as having a non-specific ID of unknown etiology. We added a description of the participants to the method section on page 3 (lines 111-123). We now state more information on the mothers and specifically detail their socio-demographic age, native language, religion, and years of education.

This article describes six dyads of mothers and adolescents (five girls and one boy) who were representative of the levels of disabilities of the broader study population; namely, two adolescents with severe disability, two adolescents with moderate disability, and two adolescents with mild disability. All the participants were diagnosed as having a non-specific ID (with unknown etiology). The IQ of the adolescents ranged from 40 to 78, based on the Stanford-Binet intelligence scales (Roid, & Pomplun, 2012) that was administered to all the participants as part of the larger study. The mothers ranged in age from 46 to 58, and the adolescents ranged in age from 16 to 20. All the mothers were Israelis. One mother was Druze. One Jewish mother self-defined as Orthodox whereas the others stated that they were secular. The mothers’ years of education ranged from 11 to 15 with a mean of 12.6. One dyad’s native language was Arabic, whereas the others were native Hebrew speakers. All names used in the text are pseudonyms, to preserve the participants’ anonymity.

-I guess that participants know the purpose of the paint. Could bias the mother's way of interactuate?

The Joint Painting Procedure is a validated tool that has been used in numerous studies over the past 10 years. We have found that even when the parents and children/adolescents know the aim of the joint painting it does not affect their genuine expression of aspects of their relationship. The implicit artistic expression invites mutual responses which bypass more conscious perceptions about relationships and enables nonconscious authentic interactions. Thus, the joint painting allows for expression beyond defense mechanisms and without bias. 

-Line 10: The new definition of ID has changed the limit of age (22 years instead of 18).

Corrected  

The sentence presented in lines 155-156 could foster the infantilization of this population.

Thank you for this important comment. This sentence was omitted from the text, and a different explanation was added on page 4 lines 171-176:

During the data collection phase (administering the JPP to the dyads) which took 12 months, the manual was adapted to adolescents with ID by the researchers including the inventor of the JPP (the first author). In this study, the qualitative analysis of the joint painting used the scales as a way to observe phenomenological phenomena for the purposes of structuring the observation and was not intended to code it numerically (Gavron & Mayseless, 2018).

Reviewer 2 Report

The paper is well written and compelling. It focuses on an under-researched challenge (parenting children with ID during adolescents) and utilizes an art-based modality to explore implicit relational patterns. 

Generally, I think the paper is good and could be published in its current shape. However, I do believe minor edits and revisions might strengthen it. Below are some of my suggestions for minor revisions:

1. The introduction and Lit review could be more systematically organized. For example, I would perhaps start with truly delving into the research related to teens who suffer from ID and their concerns - - after naming what ID is perhaps addressing demographics / prevalence, challenges with communication and relationships (eluded to and mentioned briefly), other developmental achievements and how these challenges compare and contrast with typically developing adolescents. Then I think a more thorough connection to the need for art based, implicit tools of assessments exploring relational dynamics (with other validated tools in addition to the JPP), and then a more thorough review of how the JPP is validated and what relational patterns it normally detects in teens / middle age kids. 

2. The results - especially the case illustrations seem very descriptive and not clearly linked to the 3 steps analysis described. It might be more useful to briefly chart examples of visual / relational observations for the 6 dyads (step 1), brief narratives / main points for each (step 2?) and then focus more on the integration of explicit and implicit variable (step 3?) when comparing the dyads with the 3 different levels of ID (mild, moderate, and severe)... so it reads more integrative and more applicable. It would also help readers concretely link the data to the analytic steps.

3. in the discussion and conclusion I think linking the findings to current knowledge about treatment for people coping with ID could be helpful (implied from the themes you addressed, and could be more directly linked to other research), and potentially linking  uses of community and cultural resources for families  to increase autonomy and connectivity, expression and joy...?

4. The limitation has to address the small and very purposeful sample which includes not the typical age range for adolescence (up to 20 year olds..?), not the typical validated age for the JPP (middle school kids) - although you addressed rational for this inclusion, as well as gender and ethnic / geographic  / cultural affiliation which narrows potential applicability, even when it is a qualitative study.

5. In general, I think it might be useful to consider whose reality in the dyads do you want to focus on - - are you exploring the needs of the mothers to connect and support their children (you have some research addressing their parenting style and stress) - but then the narrative examples seem to focus on the experiences / expression of the children and their responses... could maybe be a bit more consistent

All in all, the above are minor suggestions for a generally thoughtful and clear  paper. I look forward to using this paper as a resource for my students once published :)

Author Response

We thank you for your helpful and insightful comments and suggestions on the manuscript. The changes have been integrated into the revised version using track changes or are highlighted in color. We hope you will find the revised manuscript acceptable for publication.

  1. The introduction and Lit review could be more systematically organized. For example, I would perhaps start with truly delving into the research related to teens who suffer from ID and their concerns - - after naming what ID is perhaps addressing demographics/prevalence, challenges with communication and relationships (eluded to and mentioned briefly), other developmental achievements and how these challenges compare and contrast with typically developing adolescents. Then I think a more thorough connection to the need for art-based, implicit tools of assessments exploring relational dynamics (with other validated tools in addition to the JPP), and then a more thorough review of how the JPP is validated and what relational patterns it normally detects in teens / middle age kids. 

Thank you for your suggestions. The focus of our study was the relationship of adolescents with ID and their mothers. Since this is the main topic we felt it was appropriate to delve immediately into the topic of the parent-child relationship rather than discussing the broader topic of adolescents with ID, definitions, prevalence etc. There is abundant literature on adolescents with ID. Since this study deals with the specific subject of parent-adolescent relationships the introduction guides the reader to this topic directly. However, in response to your comment, we made some changes in this section for greater clarity (page 2, lines 76-80).

Over the last four decades, extensive clinical literature has pointed to the value of joint painting and drawings as a way to better understand family relationships (Landgarten, 2010; Wadeson, 2010). These instruments contribute to the evaluation of the implicit dimensions of relationships but have rarely been empirically tested (Schoch et al., 2017).

  1. The results - especially the case illustrations seem very descriptive and not clearly linked to the 3 steps analysis described. It might be more useful to briefly chart examples of visual/relational observations for the 6 dyads (step 1), brief narratives / main points for each (step 2?) and then focus more on the integration of explicit and implicit variable (step 3?) when comparing the dyads with the 3 different levels of ID (mild, moderate, and severe)... so it reads more integrative and more applicable. It would also help readers concretely link the data to the analytic steps.

Thank you for this comment. This would be an interesting avenue for future research. However,  this study focuses on the narrative as it unfolds during the interaction through verbal communication, behavior (individual and joint ), the evolvement of the painting process, and the narrative in the painting itself (where we took a phenomenological perspective). The detailed stories and the narratives represent the complexity of the dyads’ experiences.

Please see on pages 4-5 lines 199-212:

The narrative qualitative inquiry in this study focused on the narrative as it unfolds during the interaction through verbal communication,  behavior (individual and joint ), the evolvement of the painting process, and the narrative embedded in the painting itself (analyzed from a phenomenological perspective). The analysis sheds light on the complexity of the dyads’ experiences throughout all the phases of the process (Kapitan, 2018).

The analysis took place in three stages: 1. Observing the process separately for each dyad; 2. Creating and writing an inclusive narrative for each dyad based on the joint painting, the artwork, and the dyad's behavior; 3. Consolidating the insights for each dyad and integrating the explicit and implicit aspects of the relationships as they were expressed during the painting process and in the artwork into main themes and sub-themes. This was done by comparing cases to identify the central themes, similarities, and differences (Seawright & Gerring, 2008). 4. Then, the two researchers examined and discussed each theme in relation to each dyad.

  1. in the discussion and conclusion, I think linking the findings to current knowledge about treatment for people coping with ID could be helpful (implied from the themes you addressed and could be more directly linked to other research), and potentially link uses of community and cultural resources for families to increase autonomy and connectivity, expression and joy...?

Thank you for your suggestion. In the discussion, we cite studies related to the findings (including a medical clown study and a study about adolescents with ID and drama, both of which are connected to themes such as joy and autonomy. We are not sure what you meant by 'community and cultural resources' but in the final paragraph of the discussion, we strongly encourage the use of artistic activities in general and joint painting specifically as a context to assess and transform the parent-child interaction. We feel this is the main take-home message of this study.

  1. The limitation has to address the small and very purposeful sample which includes not the typical age range for adolescence (up to 20 year olds..?), not the typical validated age for the JPP (middle school kids) - although you addressed rational for this inclusion, as well as gender and ethnic / geographic  / cultural affiliation which narrows potential applicability, even when it is a qualitative study.

Thank you. We have changed the limitation section.

Please see page 14 lines 600-616

This study used the JPP, a validated tool for parents and children in middle childhood. Although the manual was adapted for parents and adolescents with ID, further validation of the JPP for this population is needed. The present study examined the interaction between mothers and adolescents, but there are other meaningful relationships in the family, such as the relationship with the father, which was found to be essential for child development no less than the relationship with the mother (Brock & Kochanska, 2015). Further research could examine the interaction between adolescents with ID with their fathers to explore implicit aspects of their relationship within the framework of the joint painting. In addition, this study examined a very small number of participants with relatively diverse levels of disability. Future studies could utilize the JPP with a larger number of dyads that would shed more light on specific levels of disability. Research could also examine gender and cultural features that could impact the specific findings reported here.

  1. In general, I think it might be useful to consider whose reality in the dyads do you want to focus on - - are you exploring the needs of the mothers to connect and support their children (you have some research addressing their parenting style and stress) - but then the narrative examples seem to focus on the experiences/expression of the children and their responses... could maybe be a bit more consistent

This is an important point. In this paper, we aimed to explore the relationship dynamics as it evolves throughout the interaction, and present the “give and take” between mother and child. We center on the adolescents’ artistic-implicit and explicit responses as part of the "here and now" interaction with their mothers. Therefore, we analyzed both the mother’s and their child’s contribution to the interaction equally.